# Local structure and distortions of mixed methane-carbon dioxide hydrates

Bernadette R. Cladek [1], S. Michelle Everett [2], Marshall T. McDonnell [3], Matthew G. Tucker [2], David J. Keffer [1] & Claudia J. Rawn [1✉]

A vast source of methane is found in gas hydrate deposits, which form naturally dispersed throughout ocean sediments and arctic permafrost. Methane may be obtained from hydrates by exchange with hydrocarbon byproduct carbon dioxide. It is imperative for the development of safe methane extraction and carbon dioxide sequestration to understand how methane and carbon dioxide co-occupy the same hydrate structure. Pair distribution functions (PDFs) provide atomic-scale structural insight into intermolecular interactions in methane and carbon dioxide hydrates. We present experimental neutron PDFs of methane, carbon dioxide and mixed methane-carbon dioxide hydrates at 10 K analyzed with complementing classical molecular dynamics simulations and Reverse Monte Carlo fitting. Mixed hydrate, which forms during the exchange process, is more locally disordered than methane or carbon dioxide hydrates. The behavior of mixed gas species cannot be interpolated from properties of pure compounds, and PDF measurements provide important understanding of how the guest composition impacts overall order in the hydrate structure.

[1] Department of Materials Science and Engineering, University of Tennessee, Knoxville, Tennessee 37996-2100, USA. [2] Neutron Scattering Division, Oak Ridge National Laboratory, Oak Ridge, Tennessee 37831-6475, USA. [3] Computer Science and Mathematics Division, Oak Ridge National Laboratory, Oak Ridge, Tennessee 37831-6475, USA. ✉email: crawn@utk.edu

Gas hydrates form in natural settings including ocean floor and sub-surface permafrost deposits when water is in the presence of a gas, such as $CH_4$, under relatively modest pressures and low temperature conditions. These non-stoichiometric compounds are a hydrogen-bonded water framework which crystallize as a clathrate cage host structure around occluded guest molecules, primarily $CH_4$[1–3]. Gas hydrates are high-density sources of $CH_4$ and accessible deposits contain an estimated 3000 trillion cubic meters of fuel[2,4]. $CH_4$ may be produced from deposits via dissociation (warming or depressurization) or exchange with $CO_2$. The latter method can potentially stabilize and maintain natural deposits, while sequestering the hydrocarbon byproduct[4,5]. As thermodynamic predictions, molecular dynamics (MD) simulations, and in situ diffraction experiments demonstrate, a gas hydrate with partial replacement of $CO_2$ for $CH_4$ forms and is more stable at higher temperature and lower pressure than a pure $CH_4$ hydrate[1,3]. This attribute led to a field study in the Alaskan North Slope which demonstrated that $CH_4$ can be collected from a hydrate deposit by exchange with $CO_2$. The field study, however, did not achieve a complete exchange and resulted in a mixed hydrate deposit[5]. Guest–host interactions of the $CH_4$, $CO_2$, and mixed $CH_4$-$CO_2$ hydrate structures need to be thoroughly deciphered, to improve predictions of how to achieve a full $CH_4$-$CO_2$ exchange and to confirm the stability and safety of the altered deposit with mixed $CH_4$-$CO_2$ or pure $CO_2$ hydrate.

Three clathrate structures have been observed for natural hydrates; however, sI hydrate (the cubic structure type) is the most relevant to the formation conditions of accessible $CH_4$ hydrate and therefore our focus in structural studies[1,3,4]. SI hydrate has a cubic $Pm\overline{3}n$ (223) crystal structure, composed of 46 hydrogen-bonded $H_2O$ molecules, which form the host lattice. This structure is made up of eight cages: two small dodecahedral and six large tetrakaidekahedral cages, which can occlude up to one guest molecule[3]. The average crystal structure of sI $CH_4$ and $CO_2$ hydrate has been well characterized with diffraction[6–11], spectroscopy[12–15], MD simulations[16–23], and density functional theory calculations[11,24–26], which suggest a high degree of disorder within the crystal. This disorder is evident in the crystallographic model of sI $CH_4$ hydrate which requires four partially occupied proton positions for each $H_2O$ molecule and many positions to represent the unbonded, freely rotating $CH_4$ or partially constrained $CO_2$ molecules[27]. The intermolecular interactions that result in structural disorder are frequently studied with pair distribution functions (PDFs) calculated from simulated atomic models[18,22,23,28–30]. Computational studies suggest that mixing guests in the hydrate structure impacts the guest–host ($CH_4$ or $CO_2$–$H_2O$), guest–guest, and host–host PDFs. $CH_4$ is found to have weaker interactions with the host, providing a more ordered lattice, whereas $CO_2$ interacts strongly with the host and other guests, leading to disorder[18,21]. Complementary local structure experiments investigating the impact of $CH_4$, $CO_2$, and their mixture on the structure and stability of gas hydrates have not been reported to the best of our knowledge. In fact, there are only three published neutron PDF experiments involving gas hydrates which demonstrated limited PDF resolution due to instrumental capabilities[31–33]. Here we present neutron PDF data for $CH_4$, $CO_2$, and mixed $CH_4$–$CO_2$ hydrates measured at 10 K using time-of-flight neutron powder diffraction. Detailed analysis of this data was achieved using combined MD and Reverse Monte Carlo (RMC) simulations to model the neutron PDF data. Our analysis provides a path to observe the short-range interactions of $CH_4$ and $CO_2$ guest molecules with their host hydrate structure and how mixing guest molecules impacts stability.

## Results

**Fitting neutron PDF data.** The interpretation of PDFs obtained from neutron scattering experiments of complex materials requires atomic-scale computational modeling. In this work, classical MD simulations of rigid molecules provide an initial structural guess, which we refine with RMC simulation to more accurately explain the PDF data. We approach this data analysis with a combination of MD relaxed models, which are optimized through RMC, which incorporates both physical constraints and experimental observation. Our goal is to provide a descriptive insight into neutron PDF data with features arising from multiple species: in this case, $CO_2$, $CH_4$, and water.

Simulated neutron PDFs are calculated from large-box (~20,000 atom) models relaxed with MD, plotted in Fig. 1a, c, e and compared with the experimental neutron PDF data. These model and data comparisons demonstrate the local disorder that is not fully described by the MD simulations of pure $CH_4$, pure $CO_2$, or mixed $CH_4$–$CO_2$ hydrate systems, as the simulated peaks are noticeably too sharp and narrow. Although the MD PDFs do predict peak locations and provide a qualitative model of the short-range order in the three hydrate systems, the peak widths and shapes do not match the neutron PDF data. They do not capture the degree of disorder, which represents differences in guest–host interactions across the compositions; however, the MD models provide a good peak location comparison with the neutron data. We implement RMC simulations to further fit this model from MD to the data with RMCProfile. This program is designed for PDF analysis of disordered crystal and amorphous structures, using chemical and physical constraints to fit a large-box model to neutron PDF data[34]. Figure 1a, c, e, demonstrate the improvement achieved when the relaxed MD model for each system is fit to the data with RMC modeling. A dramatic improvement in model to data fit is achieved using RMC for all cases: $CH_4$, $CO_2$, and mixed $CH_4$–$CO_2$ hydrates, with a small oscillating difference between the RMC model and the experimental data beyond 2 Å. Pair distances in the 2–10 Å region are the most interesting PDF features for these systems, where $CH_4$–$CO_2$ composition has the most measurable impact on guest–host and host–host interactions.

The overall lack of disorder in the MD model is the primary source of misfit in Fig. 1 for the atomic separations above 2 Å. We might suspect the rigid intramolecular potentials that were used in the MD simulations as the main culprit, but the improved fits are achieved with RMC constraints which emphasize intermolecular distance relaxations. This is seen, as the greatest RMC-data difference is below 2 Å for all three hydrate systems, where the strictest constraints were applied. We did not pursue further adjustment of the intramolecular bonds to improve this low $r$ region or the weak oscillations in the RMC/data difference from 2 to 25 Å, because this risks overfitting and pushes the limitations of the data. Physical constraints are implemented in our RMC simulations to introduce flexible intramolecular bonds, to prevent unphysical collision between atoms and to maintain ice rules for hydrogen bonding in the hydrate host lattice. Under these constraints, the RMC simulation is driven by the goodness of fit of the model to the neutron PDF data[34].

A comparison of structural snapshots from the MD and RMC simulations visualize the disorder missing in the MD relaxation. The models for all three hydrate compositions, drawn in Fig. 1b, d, f, corresponding to the MD model (bottom) and the RMC fit structure (top), show that the crystal structures are maintained while there is an increase in local disorder of atomic positions. This short-range analysis provides characterization of the local disorder in crystals which is not generally observed in characterization with Bragg diffraction. Achieving PDF analysis of these experimental data enables the investigation of the short-

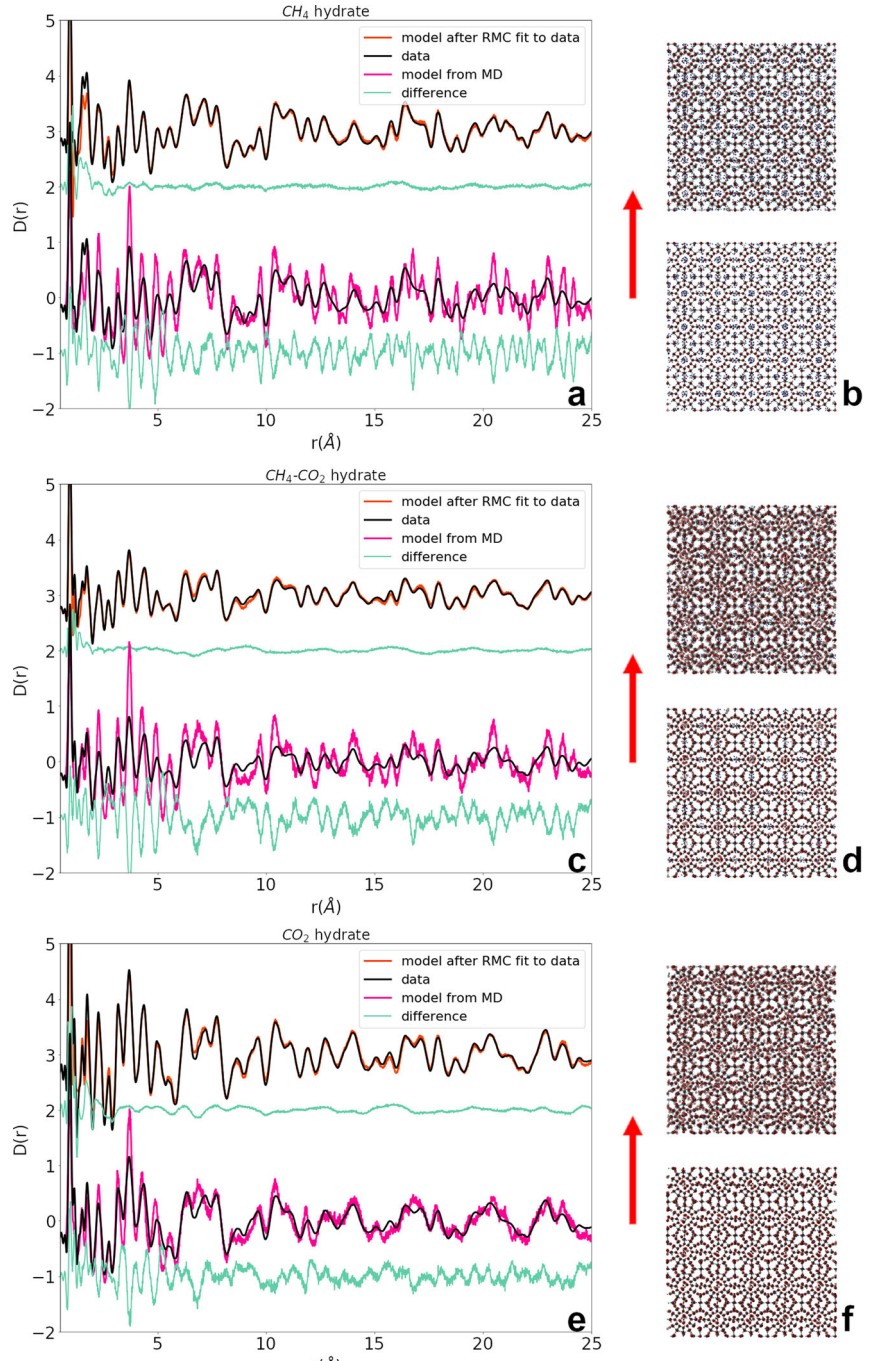

**Fig. 1 Neutron PDF data and models.** Comparison of experimental neutron PDF data with MD modeled PDF (**a**, **c**, **e** bottom spectra) and the model fit to the data with RMC (**a**, **c**, **e** top spectra) with corresponding difference patterns for $CH_4$ (**a**), $CH_4$–$CO_2$ (**c**), and $CO_2$ (**e**) hydrates. The overall disorder that RMC models (**b**, **d**, **f** top) calculate is also evident when compared with the MD structures (**b**, **d**, **f** bottom).

range interactions in these hydrates, which drives varied thermodynamic behavior across the three solid solution compositions.

**Host–host interactions**. Once a good fit to the neutron PDF is achieved, we isolate contributions from atom types by calculating radial distribution functions (RDFs) from simulations of the fit models. RDFs are calculated from MD simulations and the RMC fits for analysis of individual species pair distances in the pure $CH_4$, $CO_2$, and mixed $CH_4$–$CO_2$ hydrate structures to discern which molecular interactions define the three hydrate systems. RDFs calculated from MD simulation versus RMC fits to

experimental data highlight the features that are not captured in the MD simulation. Specifically, host–host, guest–host, and guest–guest intermolecular interactions are compared for the two guest and cage types for both $CH_4$ and $CO_2$ hydrates, and mixed $CH_4$–$CO_2$ hydrate via the RDFs calculated from RMC fits. The total neutron PDFs in Fig. 1a, c, e exhibit more disorder in the experimental models than the MD models, so the species-specific RDFs can indicate what interactions lead to that disorder.

Host–host intermolecular interactions are most clearly described by the $O_{water}$–$O_{water}$ RDFs, which are used to measure distortion in the host lattice. RDFs from the fits to neutron data and MD simulations for the pure $CH_4$, pure $CO_2$, or $CH_4$–$CO_2$

mixed hydrate samples which compare the impact of guest composition on the hydrate host lattice are shown in Fig. 2a. Structures of large and small cages selected from the large-box models are shown in Fig. 2b, c to visualize the MD and the experimental results. The RDF peaks in Fig. 2a, which measure $O_{water}$–$O_{water}$ pair distances for the $D_2O$ molecules that form hydrate cages, are more distorted in the experiment than MD models predict and are generally wider, shorter, and not normally distributed. Analysis of the experimental data shows that the first nearest neighbor $O_{water}$–$O_{water}$ peak, at ~2.8 Å, is much broader than MD simulations indicate. Instead of normally, tightly distributed distances, the O pairs distribute from 2 to 3.25 Å with an average of 2.7 Å and a defined high-$r$ shoulder at 3.2 Å. The mixed $CH_4$–$CO_2$ hydrate also has a low $r$ shoulder due to some shortening of pentagonal face edges. This detail, in combination with the smeared peak shape as $r$ increases, indicates that the hydrate lattice is the most disordered for a guest composition of mixed $CH_4$–$CO_2$. We can visualize a large and small cage extracted from experimental fits in Fig. 2c to show that mixed $CH_4$–$CO_2$ hydrate cages are the most distorted. As these

compositions follow experimental occupancy, some $CH_4$ and the mixed $CH_4$–$CO_2$ hydrate cages are vacant. The cages drawn in Fig. 2b, c were chosen as ones where the large cage is occupied by a $CO_2$ and the small cage is occupied by a $CH_4$. The imbalance of guest molecule shape and interaction potential in the mixed hydrate system, which arises from the cage filling where $CO_2$ may be surrounded by $CH_4$ or a vacancy, clearly leads to higher level of disorder in the structure.

**Guest–host Interactions.** A range of cage filling results have been predicted and observed for hydrates synthesized with $CH_4$ or $CO_2$ as the feed gas under varying pressure–temperature conditions and synthesis procedures. At 10 K, despite the larger size of the $CO_2$ molecule, hydrate with $CO_2$ as a guest has a smaller lattice parameter and $CO_2$ will preferentially fill the large cage over $CH_4$[3,7]. Local structure analysis of the guest–host RDFs in Fig. 3 lead to an understanding of how the guest molecules in different compositions interact with the host and affect cage filling, formation, stability, and decomposition behavior[3,7,18]. RDFs corresponding to the $CO_2$ molecule center (the central C) for pure $CO_2$ and mixed $CH_4$–$CO_2$ hydrates are plotted in the top row of Fig. 3 for the large cage (a) and small cage (b). The first peak for the $O_{water}$–$C_{CO2}$ pair distances is centered around ~4.5 Å for both compositions, but the mixed hydrate has a distinct split and is wider. The $CO_2$ guest in the mixed hydrate is located at more distinct positions than in the pure $CO_2$ hydrate, where the molecule is primarily centrally located in the cage. Overall, though, the guest RDF peaks in the large cage show that the local guest–host structure of $CO_2$ in mixed hydrate is more disordered. This will impact stability upon heating and suggests that host–$CO_2$ interactions in mixed hydrate are more favorable than in pure $CO_2$ hydrate. In the small cage (Fig. 3b), the position of the molecule center is essentially the same in both compositions. The $CH_4$ molecule in $CH_4$ and mixed hydrate systems occupies a tighter distribution of orientations than $CO_2$. The $O_{water}$–$C_{CH4}$ RDFs in the middle row (Fig. 3c, d) are defined by sharper, narrower peaks than in the $CO_2$ hydrate cases. Though the $CH_4$ positions are slightly less ordered in the mixed hydrate composition, the $CH_4$ guest has a lower interaction potential with the other molecules in the system, is not a polar molecule, and essentially occupies cage centers. The quadrupolar $CO_2$ occupies a more distributed position, especially in the large cage, leading to

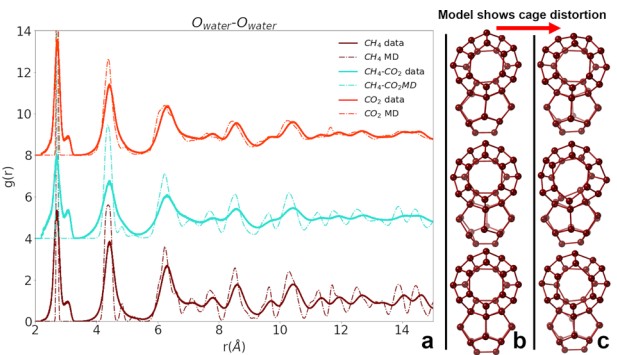

**Fig. 2 Host–host RDFs. a** RDFs calculated from MD simulations (dashed lines) and RMC fits to neutron data (solid) for the $O_{water}$–$O_{water}$ pair distances for observing the impact of guest molecule on cage distortion. **b** A large and small cage pair before RMC and **c** after RMC fit to data showing distortion in the cages as observed with neutron PDF data. The distortion of the cages is the most extreme for the mixed guest hydrate (middle row).

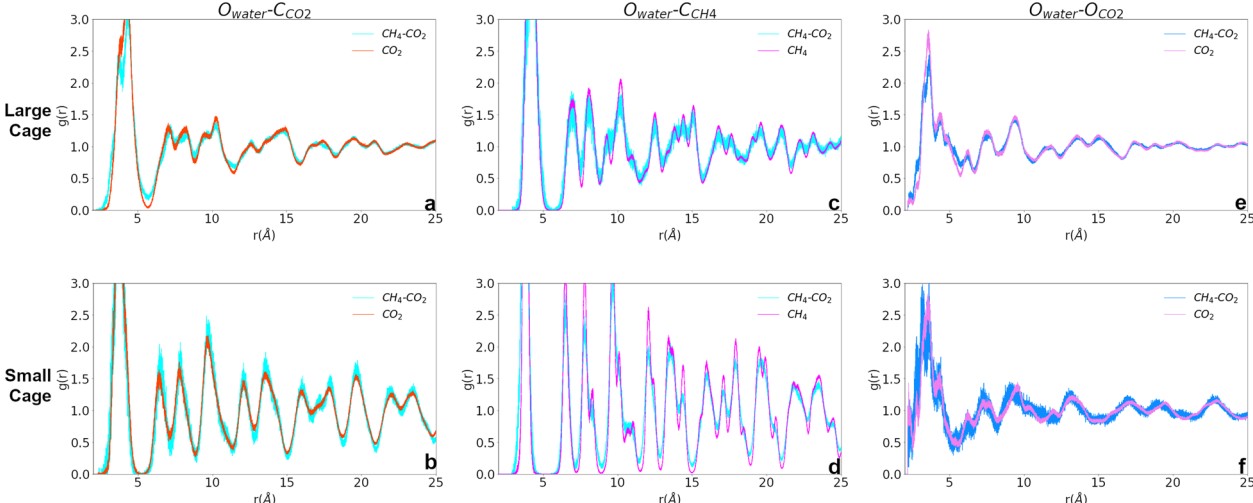

**Fig. 3 Host–guest RDFs.** RDFs to investigate guest–host interactions in three hydrate compositions. $O_{water}$–$C_{CO2}$ RDFs for the large cage (**a**) and small cage (**b**) $CO_2$ centers and $O_{water}$–$C_{CH4}$ RDFs for the large cage (**c**) and small cage (**d**) $CH_4$ centers indicate that $CO_2$ molecules are not as ordered in their positions as $CH_4$. Orientations of the linear $CO_2$ in $O_{water}$–$O_{CO2}$ RDFs in the large cage (**e**) and small cage (**f**) show that the $CO_2$ in mixed hydrate orients in more positions towards the cage.

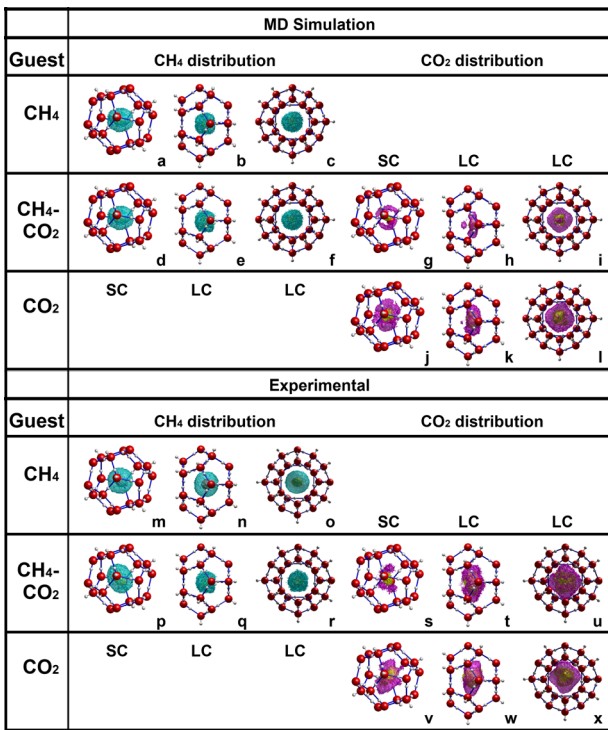

**Fig. 4 Distribution of CH₄ and CO₂ in cages.** Three-dimensional distributions, calculated from the trajectory frames from MD simulations (**a–l**) and from the sampled frames from the RMC fits (**m-x**) that were used to make the RDFs.

different interaction surfaces with its immediate host lattice $D_2O$ neighbors. Finally, as $CO_2$ is a linear molecule, the $O_{water}$–$O_{CO2}$ RDFs in the bottom row of Fig. 3e, f show that $CO_2$ in both cages approaches the $D_2O$ molecules at distances just above 2 Å, close enough to form weak hydrogen bonds which result in the distortions observed in Fig. 2 and structural stabilization. Guest–host distances across the RDFs of both compositions show that largest distortion of the local hydrate cage structure occurs in the mixed hydrate system. In this composition, $CO_2$ may be neighbored by another $CO_2$, $CH_4$, or a vacancy. We see that a change in surrounding environment allows the molecule to distribute across a broader range of positions and orientations in the cages, causing $O_{CO2}$ to interact more closely with enclathrating $D_2O$ molecules.

We can visualize and better understand the RDFs with three-dimensional density distributions calculated from MD trajectory frames and RMC fits. Orientation rotations were performed on the large cage guest molecules in order to accurately describe how they occupy the cage space with respect to the two offset hexagonal faces. Consistent with former experimental and computational results, the $CH_4$ molecule in both the MD and experimental distributions in Fig. 4a–f, m–r, respectively, orients with a spherical distribution in both the large and small cages[7,18]. Previously, the $CO_2$ molecule was shown to orient in an oblate shape parallel to the hexagonal face of the large hydrate cage, along the longer axis[7,10,24,35]. Neutron diffraction experiments showed this orientation in both pure $CO_2$ and mixed $CH_4$–$CO_2$ hydrate, but MD simulations showed that this only held for distributions in the pure $CO_2$ hydrate. MD simulations in Fig. 4h, i show $CO_2$ occupying an additional orientation towards the hexagonal face across the short axis of the cage, elaborating on previous diffraction studies where this orientation seems to be averaged out in the long-range crystallographic analysis. Neutron PDF analysis allows for direct experimental evidence of this local structure for the mixed hydrate. Figure 4t, u, w, x provide density

distributions of the $CO_2$ produced from models of the experimental data, which indicate that in mixed $CH_4$–$CO_2$ hydrate the guest in the large cage does occupy additional orientations toward the hexagonal face that do not occur in $CO_2$ hydrate. Experimental orientations are not as defined as MD predicts, as might be expected from the trend of higher disorder in experimental data seen throughout the model comparisons in previous sections. We do see, however, a low density of orientations pointing directly into the hexagonal face and greater distribution of $O_{CO2}$ positions angling toward that face than in the pure $CO_2$ composition where a narrower oblate distribution is occupied.

## Discussion

Guest–host interactions in $CH_4$, $CO_2$, and mixed $CH_4$–$CO_2$ hydrates, as observed by experimental neutron PDF analysis, are dependent on guest type and cage filling. The mixed hydrate local structure is more disordered and distorted than the pure $CH_4$ and $CO_2$ hydrates. Although these solid solution compositions follow a rule of mixtures for crystallographic properties including lattice parameter, the mixed hydrate is the outlier when the local structure is inspected. Large-box models were fit to neutron PDF data for $CH_4$, $CO_2$, and mixed $CH_4$–$CO_2$ hydrate powders, to produce RDFs and density distributions for local structure analysis. $CO_2$, when surrounded by $CH_4$ and vacant cages, has the freedom to move into a broader distribution of orientations in the hydrate cages. This is an important detail to consider when attempting a $CH_4$–$CO_2$ exchange, as a mixed hydrate will form in the intermediate steps. RDFs show that the local structure of the occluded species in the mixed hydrate clearly does not fall between that in the pure $CH_4$ and $CO_2$ hydrates. Guest molecule interactions with the host lattice cannot be expected to behave as an intermediate of pure $CH_4$ and $CO_2$ hydrate, even though predicted pressure and temperature of formation from gas and liquid components do. The altered $CO_2$–host local structure and interactions may create a free energy well that requires an interruption or altered temperature/pressure input to overcome as the disordered positions are more closely interacting with the host molecules. Our analysis approach provides a path forward to analyze neutron PDF experiments of an in situ $CH_4$–$CO_2$ exchange. Developing these studies allows for investigation into possible remedies to achieve a full $CH_4$–$CO_2$ exchange, such as a helper molecule to overcome the energy barrier[5]. Characterization of guest–host structure and interactions leads to understanding of the long-term stability of an altered natural hydrate deposit. This local characterization at 10 K shows that framework distortion in mixed $CH_4$–$CO_2$ hydrate, stabilized by $CO_2$, results in a disordered structure that inhibits further $CH_4$ removal. We demonstrate that the $CH_4$–$CO_2$ guest composition impacts the guest–host interactions in hydrates during static temperature measurements at 10 K, implying that these interactions are important in formation and decomposition. Further studies should apply these techniques during those reactions if a complete structure–property relationship is to be determined. These experimental and analysis methods have successfully presented observable differences in the hydrate compositions with $CH_4$ and $CO_2$ at 10 K. The development of these experiments and analysis at cold temperatures, where the complexity of molecular thermal motion of is minimal, makes investigations such as this possible at the higher working temperatures of a $CH_4$–$CO_2$ exchange.

## Methods

**Sample synthesis**. $CH_4$, $CO_2$, and mixed $CH_4$–$CO_2$ hydrate powders were synthesized for the neutron total scattering experiments. One milligram of *Pseudomonas syringae 31a*, branded as Snomax, was mixed with 10 mL $D_2O$ for each sample. Snomax is an ice nucleating protein and has been demonstrated to decrease

the formation pressure required for gas hydrates[35]. A 300 mL pressure reactor vessel containing the solution and steel milling media was evacuated and then pressurized at room temperature with 600 psi of gas. The three samples were pure $CO_2$ and $CH_4$ hydrates, and mixed $CH_4$–$CO_2$ hydrates, where the $CH_4/CO_2$ feed gas fraction was 0.5/0.5. For each composition after pressurization, the vessel was placed in a freezer where the temperature was dropped to 277 K and tumbled periodically for ~7 days. The freezer temperature was dropped to 253 K for at least 1 day to quench the sample after the pressure in the vessel equilibrated. Hydrate powders were collected under a nitrogen atmosphere and stored in vanadium cans in liquid nitrogen until the neutron experiments.

**Measuring neutron PDFs.** Neutron PDF data were collected from $CH_4$, $CO_2$, and mixed $CH_4$–$CO_2$ hydrate powder samples on the Nanoscale-Ordered Materials Diffractometer at the Spallation Neutron Source at Oak Ridge National Laboratory. Samples were measured at 10 K in a 50 mm He cryostat. Vanadium cans containing each sample were transferred from the liquid nitrogen storage dry shipper to the cryostat set to 90 K. The sample was then held in the cryostat at 90 K, to allow any liquid nitrogen to boil off before temperature was dropped to 2 K and increased to 10 K. Data were collected under low He pressure at 55 mbar, the working pressure for the cryostat. Neutron PDF data were collected in reciprocal space and Fourier transformed to obtain the real space function. The neutron $S(Q)s$ from 0.6-35 $Å^{-1}$ were converted in PyStoG[36] to the real space function using

$$G(r) = \frac{2}{\pi} \int_{Q_{min}}^{Q_{max}} F(Q)\sin(Qr)dQ = 4\pi r p_0 (g(r) - 1) \qquad (1)$$

with

$$F(Q) = Q(S(Q) - 1) \qquad (2)$$

and the intermolecular bond distances (greater than ~2.5 Å) are emphasized by weighting the function as

$$D(r) = 4\pi r \rho G(r) \qquad (3)$$

which is the function that was used to fit all neutron PDF data[37,38].

Measuring a neutron PDF is achieved by collecting data through a wide range of $Q$ to observe Bragg reflections and diffuse scattering, providing a powder diffraction pattern for crystal structure analysis. Rietveld refinements of the Bragg data to confirm the hydrate structure, obtain average composition and lattice parameters, and amount of ice (present as a minor secondary phase) were performed using GSAS-EXPGUI[39,40]. The details are outlined in Supplementary Table 1 and Supplementary Fig. 1, and are in agreement with the results for the same three compositions synthesized under the same conditions measured with high-resolution neutron powder diffraction at 10 K[7].

**MD simulations and PDF analysis.** Large-box models of ~20,000 atoms (depending on composition) were produced by building 5 × 5 × 5 unit cell models of the host lattice following the ideal proton configuration for sI hydrates determined by Takeuchi et al.[27]. The large and small cages were filled with $CH_4$, $CO_2$, or a vacancy according to the experimentally determined occupancies, outlined in Supplementary Table 2[7]. Two sets of models were built with this method for the three hydrate compositions using experimental lattice parameters. One set of initial large boxes were equilibrated with the LAMMPS (large-scale atomic/molecular massively parallel simulator) simulations package in the NPT (constant atoms, pressure, and temperature) and then simulated in the NVT (constant atoms, volume, and temperature) ensemble for 100 ps with a 1 fs timestep and 100 fs damping constant[41]. RDFs and density distributions were calculated from the NVT trajectories to compare with the experimental results. Fixed rigid models were used in the MD simulations for the $H_2O$, $CH_4$, and $CO_2$ molecules. The TIP4P potential model with a long-range Coulombic solver was used for the water molecules, while the $CH_4$ and $CO_2$ partial charges and Lennard Jones parameters were taken from Tse et al.[30] ($CH_4$) and Duan and Zhang[42] ($CO_2$). Finally, all intermolecular interactions for $H_2O$, $CH_4$, and $CO_2$ were determined with Lorentz–Berthelot combination rules[22]. The second set of boxes were relaxed in NVT for 100 ps at the lattice parameter and composition determined with Rietveld refinements to produce initial models for RMC fits.

Large-box models were fit to the data using RMCProfile[34]. RMC model constraints, summarized in Supplementary Table 3, result in an acceptance rate of ~10% once equilibration is reached. RDFs and density distributions were calculated from the equilibrated MD trajectories and sets of frames from RMC fits using custom codes written in FORTRAN. RMC fits for the $CH_4$, $CO_2$, and mixed $CH_4$–$CO_2$ hydrate models were each simulated for an additional 25 × 10$^6$ generated moves from 15 unique initial configurations and the resulting structures were sampled to produce a collection of ~200 frames of accepted atom configurations for each structure. We mimicked MD simulation trajectories from equilibrated RMC frames and calculated partial RDFs for individual species pair distances in the three structures. These RDFs are a probability density function, which describe the likelihood of a particle being at a distance $r$ from another particle and are calculated

as:

$$\int_0^\infty \rho g(r) 4\pi r^2 dr = N \approx N - 1 \qquad (4)$$

where $g(r)$ has the limits as $r \to \infty$, $g \to 1$[37].

## Data availability
The data that support the findings of this study are available from the corresponding author upon request.

## Code availability
The codes used in this study are primarily cited. RMCProfile version 6.7.4.3 was used to fit the neutron data and is available at RMCProfile.org. PyStog version 0.1.3 is available at https://pystog.readthedocs.io/en/latest/. The code to calculate the radial distribution function from data fits and simulations is found at http://utkstair.org/clausius/docs/mse614/text/examples.html. MD simulations were performed with LAMMPS, documented at https://lammps.sandia.gov/doc/Manual.html.

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

## Acknowledgements

B.R.C. received partial support from the University of Tennessee Chancellor's Fellowship program and the Center for Materials Processing, a Tennessee Higher Education Commission (THEC) Center of Excellence. Part of this research was sponsored by the Office of Science Graduate Student Research (SCGSR) program, administered by the Oak Ridge Institute for Science and Education for DOE. Research at the Spallation Neutron Source (SNS), a US Department of Energy (DOE) Office of Science User Facility operated by Oak Ridge National Laboratory (ORNL), was sponsored by the DOE Office of Basic Energy Sciences. The ICE-MAN software suite used for this work was funded by the ORNL Laboratory Directed Research and Development program. We thank Dayton Kizzire from University of Tennessee, Knoxville, for assisting with the neutron total scattering data collection at the SNS and Jared Floyd for assistance in sample synthesis.

## Author contributions

B.R.C. prepared the samples, performed the MD simulations, RMC fits, and data analysis. B.R.C., M.T.M., and M.G.T. performed the neutron data reduction. All authors participated in the neutron experiments at the SNS. D.J.K. wrote the codes to calculate RDFs and three-dimensional density distributions. Sample synthesis and neutron powder diffraction experiments were initiated by B.R.C. and C.J.R. This work is based on previous studies by S.M.E. and C.J.R. B.R.C. wrote the initial manuscript draft and C.J.R., D.J.K., and M.G.T. contributed to early editing. All authors contributed to, revised, and approved of the final manuscript.

## Competing interests

The authors declare no competing interests.
