## [Peer Review File · Communications Chemistry]

Reviewers' comments:

Reviewer #1 (Remarks to the Author):

This paper presents an interesting study combining neutron diffraction experiments with classical molecular dynamics and reverse Monte Carlo simulations to achieve a better understanding of structure of mixed CO₂ and CH₄ clathrate hydrates. The paper is clearly written and the conclusions are well supported by the results. Thus, this work certainly deserves publication. However, a sound discussion on the influence of the methodology on the results is clearly missing and the present version appears thus a little bit disappointing. More specifically, the following points should be taking into account before any publication can be considered:

- First, the classical simulations have been performed at 10 K, a very low temperature at which quantum effects on the clathrate structure cannot be discarded [see Conde et al., J. Chem. Phys. 2010]. Although this point is very briefly mentioned by the Authors in previous papers, it is not discussed here. The influence of these quantum effects, which mainly affect the water network, could thus influence the pair distribution function calculations as much as the use of rigid potentials does (as claimed by the Authors). In addition, the influence of quantum motions on the CH₄ molecule could also be important. These points should be discussed.

- Although the RMC method allows improvement of the simulation results with respect to experimental data, the influence of the fitting procedure behind these calculations should be also discussed. How is it realistic? What are the limitations of such an approach, when compared to direct calculations with improved interaction potentials? Fitting procedures are always suspected to give the desired results...

- Using different potential models for the guest-guest and guest-host interactions should influence the results as shown for similar systems by, i.e., Glavatskiy et al. (J. Phys. Chem. B 2012) and Ning et al. (Phys Chem Chem Phys, 2015). A comparison with simulations performed using other potentials could be achieved (or, at least, this point should be discussed). By the way, how the potential models used in the present work have been chosen? What are the main justifications?

- What is the influence of polarization effects on the results (especially on the linear, non freely rotating CO₂ molecule)?

- The clathrates simulated here are not totally filled with guest molecules (to be coherent with experimentally determined occupancies). Thus, the influence of the initial configurations (random distribution of molecules?) on the results should be discussed. Indeed, at 10 K, no molecular diffusion can be expected and, thus, the initial configuration is supposed to be stable during all the simulation runs.

For instance, in the mixed clathrates, the initial number of nearest neighbors CO₂-CH₄ pairs could influence the guest-guest interactions.

Minor point :

- in Figure 1, can we infer that $D(r)$ is given in arbitrary unit?

Reviewer #2 (Remarks to the Author):

Authors report on experimental neutron PDFs analysis of CH₄, CO₂ and mixed CH₄-CO₂ hydrates at 10 K analyzed combining classical molecular dynamics simulations and Reverse Monte Carlo fitting. The interpretation of PDFs obtained from neutron scattering experiments of complex materials requires indeed atomic scale computational modeling. In this work, authors use classical

MD simulations of rigid molecules to provide an initial structural guess, which then they fit with Reverse Monte Carlo (RMC) simulation to more accurately explain the PDF data. It is clear from the comparison of classical MD simulations and PDF data that the MD simulation does not fully describe the local disorder in the pure CH₄, pure CO₂, or mixed CH₄-CO₂ hydrate systems, as the simulated peaks are noticeably sharper and narrower than the experimental ones. Implementing RMC simulations for further fitting with RMC Profile provide a better reproduction of the experimental results.

From this analysis authors conclude that the mixed hydrate is more locally disordered than CH₄ or CO₂ hydrates. They also argue that the behavior of mixed gas species cannot be interpolated from properties of pure compounds.

I think that the presented experimental data are interesting, the overall analysis is correct, and the system itself is an appealing one.

However the used MD potential used seems badly defined not only for the mixed hydrate but also for CO₂ clathrate, probably because CO₂ is a polar molecule and water polarizability is not taken into account by standard potentials. Authors should discuss this point.

Furthermore, almost no initial characterization of the different hydrates is reported. In particular the average filling ratio (which in simulations is assumed to be 1 for both small and large cages) should be characterized either by Raman experiments or neutron diffraction.

Finally, concerning the CO₂-CH₄ exchange, which seems to be the main drive of this study, actually no much insights is finally derived from this analysis. Authors should strengthen this point.

Reviewer #3 (Remarks to the Author):

General Comments:

Cladek et al. have discussed the structural insights for pure CH₄, CO₂, and mixed CH₄-CO₂ hydrates, obtained by classical molecular dynamics (MD) simulations and Reverse Monte Carlo (RMC) fitting along with experimental neutron PDF analysis. The results show crucial structural aspects, especially for CH₄-CO₂ mixed hydrate, which gives a better understanding of the particular system, important for CH₄-CO₂ exchange by hydrate. The paper is well written. It is publishable after consideration of the following comments.

Comments:

1. Why was this study done only at 10 K, not any other higher temperatures?
2. The authors have used Snomax (*Pseudomonas syringae* 31a), which is an ice-nucleating protein to decrease the formation pressure required for gas hydrates. I would like to know whether the presence of such biological species has any effect on the gas hydrate structures.
3. What is the reason for the maximum distortion of host cages of CH₄-CO₂ mixed hydrate after RMC fit?
4. It looks like only CO₂ plays a major role in distortions of the hydrate cages (Fig. 4). What is the reason behind this phenomenon?
5. The authors should mention the type of clathrate hydrate structure and the number of water molecules in cages (due to this clarity is missing).
6. The authors have emphasized that this work will provide structural insights of CH₄-CO₂ mixed hydrate, which will help to understand the details of the exchange of CH₄-CO₂ in hydrate form. However, there is a lack of more information on how these results can be applied to the real hydrates found in nature.

Minor Comments:

1. The reference style is not uniform and should be appropriately aligned.

Dear Reviewers:

Thank you for taking the time to read our manuscript “Local Structure and Distortions of Mixed CH₄-CO₂ hydrates”. We greatly appreciate your suggestions and believe they have improved the quality and readability of our manuscript. These suggestions have now been addressed in the revised manuscript.

We believe that your most critical points resulted from a lack of clarity in our initial manuscript. Below in our responses, we have addressed each concern and pointed to where these are discussed in the revised manuscript along with edits to clarify points that were missed.

Below, we provide our itemized response to your comments. The reviewer comments are written in italics, and our responses are written in normal font, preceded by the phrase “Authors’ Reply”. We have noted the page numbers where we have made the corrections and additions to the manuscript. A marked up revision is provided that is a version of the revised manuscript with the points discussed below highlighted. Supplementary information is also provided after the references in the manuscript, marked up version, and in a separate document. We have also reproduced the changes to the manuscript in this response. In conclusion, we hope you will find our responses and revised manuscript suitable for publication in *Communications Chemistry*.

Sincerely,

Bernadette R. Cladek

Graduate Research Assistant, Materials Science and Engineering

bcladek@utk.edu

1508 Middle Drive

414 Ferris Hall

Knoxville, TN 37996

**corresponding author:* Claudia J. Rawn

Associate Professor, Materials Science and Engineering

1508 Middle Drive

331 Ferris Hall

Knoxville, TN 37996

crawn@utk.edu

(865) 974-5340

Responses to Reviewers

Below, we provide our itemized response to the reviewer's comments. The reviewer's comments are written in italics, and our responses are written in normal font, preceded by the phrase "Authors' Reply". We have noted the page numbers where we have made the revisions and additions to the manuscript. We have reproduced the changes in the manuscript in this response.

Reviewer #1 (Remarks to the Author):

This paper presents an interesting study combining neutron diffraction experiments with classical molecular dynamics and reverse Monte Carlo simulations to achieve a better understanding of structure of mixed CO₂ and CH₄ clathrate hydrates. The paper is clearly written and the conclusions are well supported by the results. Thus, this work certainly deserves publication. However, a sound discussion on the influence of the methodology on the results is clearly missing and the present version appears thus a little bit disappointing. More specifically, the following points should be taking into account before any publication can be considered:

- First, the classical simulations have been performed at 10 K, a very low temperature at which quantum effects on the clathrate structure cannot be disregarded [see Conde et al., J. Chem. Phys. 2010]. Although this point is very briefly mentioned by the Authors in previous papers, it is not discussed here. The influence of these quantum effects, which mainly affect the water network, could thus influence the pair distribution function calculations as much as the use of rigid potentials does (as claimed by the Authors). In addition, the influence of quantum motions on the CH₄ molecule could also be important. These points should be discussed.

Authors' Reply : Accounting for the quantum effects in H₂O and CH₄ in simulations is computationally costly, and in planning this work we decided it is not necessary if the classical model can successfully be fit to the data with RMC fitting. We recognize that attempting to reproduce properties such as density and CH₄ motion, especially at low temperature, for these systems would require more considerations, but the MD step in this work is a means to achieve a good starting model for RMC fitting. If the MD simulations provided initial guesses that either lost connectivity, broke the enforced constraints, or did not converge to a good fit in the RMC simulation, we would have looked to incorporate other potentials in the MD step.

Previous work by our group used classical simulations with TIP4P water potentials and treated the CH₄ classically at 10 K (Cladek, B. *et al.* Guest-Host Interactions in Mixed CH₄-CO₂ Hydrates: Insights from Molecular Dynamic Simulations. *The Journal of Physical Chemistry C* **122**, 19575-19583 (2018).). These methods do not produce experimental density

of hydrates and are not good to model the dynamics of CH₄ at low temperatures. However, these simulations were shown to provide a good qualitative comparison of large, statistically relevant CH₄, CO₂, and mixed CH₄-CO₂ hydrates for comparative PDF and energetic analysis with respect to CH₄-CO₂ guest composition. Therefore, we used the same method to produce initial models for fitting to neutron PDF data with RMC simulations. This data analysis requires a large box and many frames to achieve the statistics necessary to break down the partial atomic species contributions. This is especially important in the mixed system where we are modeling lower cage occupancies as CH₄ and CO₂ are both present. While it would be interesting to compare potentials which are better suited to recreating experimental density, it is moot as we must set the density to the experimentally determined value from crystal structure refinements. Here, we aim to produce an appropriate method to analyze neutron PDF data from CH₄-CO₂ hydrates with large box models. The MD simulations provide a good, relaxed starting model for the RMC. This is evidenced by the success of the RMC simulations to fit the data despite our very ridged constraints.

To emphasize that the MD step is an important, but initial step in the process to model neutron PDF data, we have revised the manuscript on page 4 with clarification:

“The interpretation of PDFs obtained from neutron scattering experiments of complex materials requires atomic scale computational modeling. In this work, classical molecular dynamics (MD) simulations of rigid molecules provide an initial structural guess, which we fit with Reverse Monte Carlo (RMC) simulation to more accurately explain the PDF data. We approach this data analysis with a combination of MD relaxed models which are optimized through RMC, which incorporates both physical constraints and experimental observation. Our goal is to provide a descriptive insight into neutron PDF data with features arising from multiple species, in this case, CO₂, CH₄ and water.”

- Although the RMC method allows improvement of the simulation results with respect to experimental data, the influence of the fitting procedure behind these calculations should be also discussed. How is it realistic ? What are the limitations of such an approach, when compared to direct calculations with improved interaction potentials ? Fitting procedures are always suspected to give the desired results...

Authors' Reply : The RMC method presented in this work improves the MD produced models by fitting them to the neutron PDF data. These simulations are driven by goodness of fit of the model to the data. This is analogous to the exergy that would be minimized in a more traditional atomistic MC simulation. Instead of energy, this method minimizes the metric of fit to the data which is χ^2 .

We have revised the manuscript on page 6 to clarify this point:

“Physical constraints are implemented in our RMC simulations to introduce flexible intramolecular bonds, to prevent unphysical collision between atoms, and to maintain ice rules for hydrogen-bonding in the hydrate host lattice. Under these constraints, the RMC simulation is driven by the goodness of fit of the model to the neutron PDF data.³⁴”

- Using different potential models for the guest-guest and guest-host interactions should influence the results as shown for similar systems by, i.e., Glavatskiy et al. (J. Phys. Chem. B 2012) and Ning et al. (Phys Chem Chem Phys, 2015). A comparison with simulations performed using other potentials could be achieved (or, at least, this point should be discussed).

By the way, how the potential models used in the present work have been chosen ? What are the main justifications ?

Authors' Reply: The potentials chosen, also used in previous work, showed good qualitative comparison while allowing the production of long simulations of large systems. Our initial work cited (Jiang, H. & Jordan, K. D. Comparison of the Properties of Xenon, Methane, and Carbon Dioxide Hydrates from Equilibrium and Nonequilibrium Molecular Dynamics Simulations†. *The Journal of Physical Chemistry C* **114**, 5555-5564 (2010).) when choosing these potentials. RDFs and density distributions were first studies with these potentials in (Cladek, B. R. *et al.* Guest-Host Interactions in Mixed CH₄- CO₂ Hydrates: Insights from Molecular Dynamic Simulations. *The Journal of Physical Chemistry C* **122**, 19575-19583 (2018).). The results from this work, which was a qualitative comparison of local structure and energetic analysis in CH₄-CO₂ hydrates based on experimentally determined compositions, pointed to the importance of expanding experimental studies from long range characterization to short range structural characterization (neutron PDF). For the neutron PDF data analysis in this manuscript, it made sense to use the same simulation methods to build our initial models for RMC fitting for comparison with the previous work. While we recognize that including effects like quantum corrections to TIP4P potentials and CH₄, the MD simulations in this work are primarily a method to create a quality relaxed starting model to fit to the neutron data in RMC. If the MD simulations provided initial guesses that either lost connectivity, broke the enforced constraints, or did not converge to a good fit in the RMC simulation, we would have looked to incorporate other potentials in the MD step.

On pages 4-5 in the manuscript we have revised sections to reiterate that there is mismatch seen in the MD models, but this step is a part of the data fitting process, necessary to create a large box starting model for RMC:

“In this work, classical molecular dynamics (MD) simulations of rigid molecules provide an initial structural guess, which we refine with Reverse Monte Carlo (RMC) simulation to more accurately explain the PDF data. We approach this data analysis with a combination of MD relaxed models which are optimized through RMC, which incorporates both physical

constraints and experimental observation. Our goal is to provide a descriptive insight into neutron PDF data with features arising from multiple species, in this case, CO₂, CH₄ and water.

Simulated neutron PDFs are calculated from large-box (~20,000 atom) models relaxed with MD, plotted in Figure 1 a, c, and e, and compared with the experimental neutron PDF data. These model and data comparisons demonstrate the local disorder that is not fully described by the MD simulations of pure CH₄, pure CO₂, or mixed CH₄-CO₂ hydrate systems, as the simulated peaks are noticeably too sharp and narrow. Although the MD PDFs do predict peak locations and provide a qualitative model of the short-range order in the three hydrate systems, the peak widths and shapes do not match the neutron PDF data. They do not capture the degree of disorder, which represents differences in guest-host interactions across the compositions, however, the MD models provide a good peak location comparison with the neutron data. We implement RMC simulations to further refine this model from MD to the data with RMCProfile.)

- What is be the influence of polarization effects on the results (especially on the linear, non freely rotating CO2 molecule) ?

Authors' Reply: Polarizable models are computationally costly. Our goal in this work was to us MD simulations to produce a relaxed model to be fit to neutron PDF data with RMC simulations. We found that starting from a non-simulated MD structure was too far from the actual structure to converge to a good fit, but the MD structure provides a good starting place. We do use a long range Coulombic solver on the linear molecule with partial charges.

- The clathrates simulated here are not totally filled with guest molecules (to be coherent with experimentally determined occupancies). Thus, the influence of the initial configurations (random distribution of molecules ?) on the results should be discussed. Indeed, at 10 K, no molecular diffusion can be expected and, thus, the initial configuration is supposed to be stable during all the simulation runs.

For instance, in the mixed clathrates, the initial number of nearest neighbors CO₂-CH₄ pairs could influence the guest-guest interactions.

Authors' Reply: The guest occupancies followed the experimentally determined large and small cage occupancies for CH₄ and CO₂. After following these rules, the cages were filled randomly and the molecules were placed in random orientations in the cage centers. If there were a large influence of guest molecule clustering, it would be apparent in the fit of the model to the data during the RMC simulation.

The use of experimentally determined cage filling is addressed on page 18:

“The large and small cages were filled with CH₄, CO₂, or a vacancy according to the experimentally determined occupancies, outlined in the supplementary materials.⁷”

Additionally, below is the SI Table II which is referenced:

SI Table II. Hydrate cage occupancies high resolution neutron powder diffraction (POWGEN) in previous work.¹

Feed gas composition	Large cage occupancy		Small cage occupancy		Total composition %		Total cages filled %
	CH ₄	CO ₂	CH ₄	CO ₂	CH ₄	CO ₂	
100% CH₄	0.73(3)	-	0.93(5)	-	79(4)	-	79(4)
50% CH₄ 50% CO₂	0.08(3)	0.77(3)	0.54(4)	0.21(2)	20(3)	63(1)	83(5)
100% CO₂	-	1.00(6)	-	1.00(4)	-	100(6)	100(6)

Minor point :

- in Figure 1, can we infer that D(r) is given in arbitrary unit ?

Authors' Reply: D(r) units are arbitrary, thus the figure caption has been modified to clarify this.

Reviewer #2 (Remarks to the Author):

Authors report on experimental neutron PDFs analysis of CH₄, CO₂ and mixed CH₄-CO₂ hydrates at 10 K analyzed combining classical molecular dynamics simulations and Reverse Monte Carlo fitting. The interpretation of PDFs obtained from neutron scattering experiments of complex materials requires indeed atomic scale computational modeling. In this work, authors use classical MD simulations of rigid molecules to provide an initial structural guess, which then they fit with Reverse Monte Carlo (RMC) simulation to more accurately explain the PDF data. It is clear from the comparison of classical MD simulations and PDF data that the MD simulation does not fully describe the local disorder in the pure CH₄, pure CO₂, or mixed CH₄-CO₂ hydrate systems, as the simulated peaks are noticeably sharper and narrower than the experimental ones. Implementing RMC simulations for further fitting with RMC Profile provide a better reproduction of the experimental results. From this analysis authors conclude that the mixed hydrate is more locally disordered than CH₄ or CO₂ hydrates. They also argue that the behavior of mixed gas species cannot be

interpolated from properties of pure compounds.

I think that the presented experimental data are interesting, the overall analysis is correct, and the system itself is an appealing one.

However the used MD potential used seems badly defined not only for the mixed hydrate but also for CO₂ clathrate, probably because CO₂ is a polar molecule and water polarizability is not taken into account by standard potentials. Authors should discuss this point.

Authors' Reply: Polarizable models are computationally costly. Our goal in this work was to use MD simulations to produce a relaxed model to be refined to neutron PDF data with RMC simulations. We found that starting from a non-simulated MD structure was too far from the actual structure to converge to a good fit, but the MD structure provides a good starting place. We do use a long range Coulombic solver on the linear molecule with partial charges.

Furthermore, almost no initial characterization of the different hydrates is reported. In particular the average filling ratio (which in simulations is assumed to be 1 for both small and large cages) should be characterized either by Raman experiments or neutron diffraction.

Authors' Reply: We apologize for lack of clarity in the manuscript and it is edited to make clear that this work was completed. Lattice parameters and phase fractions were determined on the neutron PDF data using Rietveld refinements in GSAS. Cage filling ratios for CH₄ and CO₂ in the large and small cages was determined with neutron powder diffraction in previous experiments (Everett et. al. 2015). This cage filling was accounted for in the building of the models and all calculated neutron scattering densities.

The use of experimentally determined lattice parameters and cage filling is addressed on page 18:

“The large and small cages were filled with CH₄, CO₂, or a vacancy according to the experimentally determined occupancies, outlined in the supplementary materials.⁷ Two sets of models were built with this method for the three hydrate compositions using experimental lattice parameters.”

Additionally, below is the SI Tables I and II which is referenced, showing the lattice parameters and cage filling:

SI Table I. Lattice parameters from the Rietveld refinements and hydrate/ice phase fractions from the Bragg patterns obtained from the total scattering instrument (NOMAD) and from high resolution neutron powder diffraction (POWGEN) in previous work.

Lattice parameter (Å)	Phase fraction ice
--------------------

Feed gas composition	NOMAD (this work)	POWGEN (Everett et al. ¹)	NOMAD (this work)
100% CH ₄	11.830(2)	11.83210(8)	0.10(1)
50% CH ₄ 50% CO ₂	11.8244(3)	11.82487(8)	0.21(1)
100% CO ₂	11.8192(7)	11.82216(9)	0.13(1)

SI Table II. Hydrate cage occupancies high resolution neutron powder diffraction (POWGEN) in previous work.¹

Feed gas composition	Large cage occupancy		Small cage occupancy		Total composition %		Total cages filled %
	CH ₄	CO ₂	CH ₄	CO ₂	CH ₄	CO ₂	
100% CH ₄	0.73(3)	-	0.93(5)	-	79(4)	-	79(4)
50% CH ₄ 50% CO ₂	0.08(3)	0.77(3)	0.54(4)	0.21(2)	20(3)	63(1)	83(5)
100% CO ₂	-	1.00(6)	-	1.00(4)	-	100(6)	100(6)

Finally, concerning the CO₂-CH₄ exchange, which seems to be the main drive of this study, actually no much insights is finally derived from this analysis. Authors should strengthen this point.

Authors' Reply: This discussion has been revised for clarity on pages 15-16:

“The altered CO₂-host local structure and interactions may create a free energy well that requires an interruption or altered temperature/pressure input to overcome as the disordered positions are more closely interacting with the host molecules. Our analysis approach provides a path forward to analyze neutron PDF experiments of an in-situ CH₄- CO₂ exchange. Developing these studies allows for investigation into possible remedies to achieve a full CH₄- CO₂ exchange, such as a helper molecule to overcome the energy barrier.⁵ Characterization of guest-host structure and interactions leads to understanding of the long-term stability of an altered natural hydrate deposit. This local characterization at 10 K shows that framework distortion in mixed CH₄- CO₂ hydrate, stabilized by CO₂, results in a disordered structure that inhibits further CH₄ removal. We demonstrate that the CH₄- CO₂ guest composition impacts the guest-host interactions in hydrates during static temperature measurements at 10 K, implying that these interactions are important in formation and decomposition. Further studies should apply these techniques during those reactions if a complete structure-property relationship is to be determined.”

Reviewer #3 (Remarks to the Author):

General Comments:

Cladek et al. have discussed the structural insights for pure CH₄, CO₂, and mixed CH₄-CO₂ hydrates, obtained by classical molecular dynamics (MD) simulations and Reverse Monte Carlo (RMC) fitting along with experimental neutron PDF analysis. The results show crucial structural aspects, especially for CH₄-CO₂ mixed hydrate, which gives a better understanding of the particular system, important for CH₄-CO₂ exchange by hydrate. The paper is well written. It is publishable after consideration of the following comments.

Comments:

1. Why was this study done only at 10 K, not any other higher temperatures?

Authors' Reply: The importance of this study is to present our developed technique for analyzing neutron PDF data with MD models fit to the data using large box RMC simulation. This experiment focuses on 10 K to compare with our previous work. First, Everett et al. analyzed the crystal structure of these samples with high resolution neutron powder diffraction. Second, we modeled the results from the crystallographic study with MD simulations for a qualitative composition comparison at 10 K, and to elaborate on the NPD results. Low temperatures are necessary to develop this analysis method as thermal effects on the sample and neutron scattering complicates the data. Therefore, we chose 10 K to develop this technique where we have already published NPD and MD studies on these samples. This large undertaking makes analysis of studies at warmer temperatures possible.

2. The authors have used Snomax (Pseudomonas syringae 31a), which is an ice-nucleating protein to decrease the formation pressure required for gas hydrates. I would like to know whether the presence of such biological species has any effect on the gas hydrate structures.

Authors' Reply: SNOMAX is used in a very small concentration (1:10,000), and in high resolution neutron powder diffraction it had no impact on the hydrate crystal structure (Everett, S. M. et al. Insights into the structure of mixed CO₂/CH₄ in gas hydrates. American Mineralogist **100**, 1203-1028 (2015).).

3. What is the reason for the maximum distortion of host cages of CH₄-CO₂ mixed hydrate after RMC fit?

Authors' Reply: As discussed in the results, the distortion is due to the CO₂ being neighbored by either CH₄ or vacant cages, compared to the pure CO₂ hydrate where full occupancy was observed. The lack of CO₂ support in neighboring cages allows the CO₂ to interact more strongly with the cage.

We have revised the manuscript on page 9 to emphasize this point:

“The imbalance of guest molecule shape and interaction potential in the mixed hydrate system, which arises from the cage filling where CO₂ may be surrounded by CH₄ or a vacancy, clearly leads to higher level of disorder in the structure.”

4. It looks like only CO₂ plays a major role in distortions of the hydrate cages (Fig. 4). What is the reason behind this phenomenon?

Authors’ Reply: CH₄ is a tetrahedral molecule which we observe to orient in a symmetric, spherical orientation in both cage types. It has also been shown to have lower interaction energy with the hydrate lattice than CO₂. The CO₂-H₂O interactions are higher than CH₄, therefore CO₂ distorts the cages. It is also shown in this work and in published work that at these temperature, CO₂ hydrate has a smaller lattice parameter, bringing the guest and host molecules closer and leading to higher interactions.

5. The authors should mention the type of clathrate hydrate structure and the number of water molecules in cages (due to this clarity is missing).

Authors’ Reply: Thank you for bringing this omission to our attention. Details of the crystal structure are found on page 3:

“SI hydrate has a cubic $Pm\bar{3}n$ (223) crystal structure, composed of forty-six hydrogen bonded H₂O molecules which form the host lattice. This structure is made up of eight cages; two small dodecahedral and six large tetrakaidekahedral cages, which can occlude up to one guest molecule.³”

The total number of water molecules in the modeled systems are discussed on page 18:

“Large-box models of ~20,000 atoms (depending on composition) were produced by building 5 x 5 x 5 unit cell models of the host lattice following the ideal proton configuration for SI hydrates determined by Takeuchi et al.²⁷”

6. The authors have emphasized that this work will provide structural insights of CH₄-CO₂ mixed hydrate, which will help to understand the details of the exchange of CH₄-CO₂ in hydrate form. However, there is a lack of more information on how these results can be applied to the real hydrates found in nature.

Authors’ Reply: This discussion has been revised for clarity on pages 15-16:

“The altered CO₂-host local structure and interactions may create a free energy well that requires an interruption or altered temperature/pressure input to overcome as the disordered positions are more closely interacting with the host molecules. Our analysis approach

provides a path forward to analyze neutron PDF experiments of an in-situ CH₄- CO₂ exchange. Developing these studies allows for investigation into possible remedies to achieve a full CH₄- CO₂ exchange, such as a helper molecule to overcome the energy barrier.⁵ Characterization of guest-host structure and interactions leads to understanding of the long-term stability of an altered natural hydrate deposit. This local characterization at 10 K shows that framework distortion in mixed CH₄- CO₂ hydrate, stabilized by CO₂, results in a disordered structure that inhibits further CH₄ removal. We demonstrate that the CH₄- CO₂ guest composition impacts the guest-host interactions in hydrates during static temperature measurements at 10 K, implying that these interactions are important in formation and decomposition. Further studies should apply these techniques during those reactions if a complete structure-property relationship is to be determined.”

Minor Comments:

1. The reference style is not uniform and should be appropriately aligned.

Authors' Reply: Thank You. We have checked that all references are uniform.

REVIEWERS' COMMENTS:

Reviewer #1 (Remarks to the Author):

In this revised version, the Authors have properly addressed the comments of the Reviewers. Therefore, the manuscript has been improved and appears now suitable for publication. However, there is one additional point that has to be discussed in the final version, before the paper is accepted : Indeed, the Authors investigated the influence of the guest molecules on the hydrate structure, in connection with CH₄ extraction and concomitant CO₂ sequestration. However, the applications of the CH₄/CO₂ exchange process are expected at much higher temperature than the one considered here (10 K) and thus, the transferability of the present conclusions for these more realistic temperatures has to be discussed.

Reviewer #2 (Remarks to the Author):

Authors replied to my comments in pertinent way. I do support publication of the manuscript.

Reviewer #3 (Remarks to the Author):

Editorial note: This reviewer provided no further comments for the authors.

Dear Reviewer:

Thank you for taking the time to read our revised manuscript “Local structure and distortions of mixed methane and carbon dioxide hydrates”. We greatly appreciate your suggestions and have addressed your final comment.

Below, we provide our response to your comment. The reviewer comment is written in italics, and our response is written in normal font, preceded by the phrase “Authors’ Reply”. We have noted the page number where we have made the corrections and additions to the manuscript. We have also reproduced the changes to the manuscript in this response. We hope you will find our response and revised manuscript suitable for publication in *Communications Chemistry*.

Sincerely,

Bernadette R. Cladek

Graduate Research Assistant, Materials Science and Engineering

bcladek@utk.edu

1508 Middle Drive

414 Ferris Hall

Knoxville, TN 37996

**corresponding author:* Claudia J. Rawn

Associate Professor, Materials Science and Engineering

1508 Middle Drive

331 Ferris Hall

Knoxville, TN 37996

crawn@utk.edu

(865) 974-5340

Responses to Reviewers

Below, we provide our itemized response to the reviewer’s comments. The reviewer’s comments are written in italics, and our responses are written in normal font, preceded by the phrase “Authors’ Reply”. We have noted the page numbers where we have made the revisions and additions to the manuscript. We have reproduced the changes in the manuscript in this response.

Reviewer #1 (Remarks to the Author):

In this revised version, the Authors have properly addressed the comments of the Reviewers.

Therefore, the manuscript has been improved and appears now suitable for publication.

However, there is one additional point that has to be discussed in the final version, before the paper is accepted : Indeed, the Authors investigated the influence of the guest molecules on the hydrate structure, in connection with CH₄ extraction and concomitant CO₂ sequestration. However, the applications of the CH₄/CO₂ exchange process are expected at much higher temperature than the

one considered here (10 K) and thus, the transferability of the present conclusions for these more realistic temperatures has to be discussed.

Authors' Reply: We recognize that 10 K is much cooler than the working temperatures of a CH₄-CO₂ exchange (~273 K), but the cold temperature was necessary to develop this analysis technique, test it, and prove that the local structure of three hydrate compositions can be measured in this way. Analysis at higher temperatures is certainly the next step, and the 10 K analysis acts as a baseline for determining properties at ~273 K. The importance of this study is to present our developed technique for analyzing neutron PDF data with MD models fit to the data using large box RMC simulation. This experiment focuses on 10 K to compare with related previous work. First, Everett et al. analyzed the crystal structure of these samples with high resolution neutron powder diffraction. Second, we modeled the results from the crystallographic study with MD simulations for a qualitative composition comparison at 10 K, and to elaborate on the NPD results. Low temperatures are necessary to develop this analysis method as thermal effects on the sample and neutron scattering complicates the data. Therefore, we chose 10 K to develop this technique where we have already published NPD and MD studies on these samples. This large undertaking makes analysis of studies at warmer temperatures possible. We added to our discussion to make it clear to the reader that these experiments are important at cold temperatures, but future work needs to achieve increase the experimental temperature to understand local structure properties the appropriate exchange environment.

Our revisions to address this can be found in the Discussions section on page 16 of the revised manuscript:

“This local characterization at 10 K shows that framework distortion in mixed CH₄- CO₂ hydrate, stabilized by CO₂, results in a disordered structure that inhibits further CH₄ removal. We demonstrate that the CH₄- CO₂ guest composition impacts the guest-host interactions in hydrates during static temperature measurements at 10 K, implying that these interactions are important in formation and decomposition. Further studies should apply these techniques during those reactions if a complete structure-property relationship is to be determined. These experimental and analysis methods have successfully presented observable differences in the hydrate compositions with CH₄ and CO₂ at 10 K. The development of these experiments and analysis at cold temperatures, where the complexity of molecular thermal motion of is minimal, makes investigations such as this possible at the higher working temperatures of a CH₄-CO₂ exchange.